American Society for Microbiology
# Antibiotic Tolerance Indicative of Persistence Is Pervasive among Clinical *Streptococcus pneumoniae* Isolates and Shows Strong Condition Dependence

Nele Geerts,[a] Linda De Vooght,[a] Ioannis Passaris,[b] Peter Delputte,[a] Bram Van den Bergh,[c,d] Paul Cos[a]

aLaboratory for Microbiology, Parasitology and Hygiene (LMPH), Wilrijk, Belgium
bSciensano, Bacterial Diseases Unit, Brussels, Belgium
cCentre of Microbial and Plant Genetics, Department of Molecular and Microbial Systems, KU Leuven, Leuven, Belgium
dCenter for Microbiology, Flanders Institute for Biotechnology, VIB, Leuven, Belgium

Bram Van den Bergh and Paul Cos contributed equally as senior authors. Author order was determined in order of increasing seniority.

**ABSTRACT** *Streptococcus pneumoniae* is an important human pathogen, being one of the most common causes of community-acquired pneumonia and otitis media. Antibiotic resistance in *S. pneumoniae* is an emerging problem, as it depletes our arsenal of effective drugs. In addition, persistence also contributes to the antibiotic crisis in many other pathogens, yet for *S. pneumoniae*, little is known about antibiotic-tolerant persisters and robust experimental means are lacking. Persister cells are phenotypic variants that exist as a subpopulation within a clonal culture. Being tolerant to lethal antibiotics, they underly the chronic nature of a variety of infections and even help in acquiring genetic resistance. In this study, we set out to identify and characterize persistence in *S. pneumoniae*. Specifically, we followed different strategies to overcome the self-limiting nature of *S. pneumoniae* as a confounding factor in the prolonged monitoring of antibiotic survival needed to study persistence. Under optimized conditions, we identified genuine persisters in various growth phases and for four relevant antibiotics through biphasic survival dynamics and heritability assays. Finally, we detected a high variety in antibiotic survival levels across a diverse collection of *S. pneumoniae* clinical isolates, which assumes that a high natural diversity in persistence is widely present in *S. pneumoniae*. Collectively, this proof of concept significantly progresses the understanding of the importance of antibiotic persistence in *S. pneumoniae* infections, which will set the stage for characterizing its relevance to clinical outcomes and advocates for increased attention to the phenotype in both fundamental and clinical research.

**IMPORTANCE** *S. pneumoniae* is considered a serious threat by the Centers for Disease Control and Prevention because of rising antibiotic resistance. In addition to resistance, bacteria can also survive lethal antibiotic treatment by developing antibiotic tolerance, more specifically, antibiotic tolerance through persistence. This phenotypic variation seems omnipresent among bacterial life, is linked to therapy failure, and acts as a catalyst for resistance development. This study gives the first proof of the presence of persister cells in *S. pneumoniae* and shows a high variety in persistence levels among diverse strains, suggesting that persistence is a general trait in *S. pneumoniae* cultures. Our work advocates for higher interest for persistence in *S. pneumoniae* as a contributing factor for therapy failure and resistance development.

**KEYWORDS** *Streptococcus pneumoniae*, autolysis, persistence, antibiotics, tolerance

Address correspondence to Bram Van den Bergh, bram.vandenbergh@kuleuven.be, or Paul Cos, paul.cos@uantwerpen.be.

The authors declare no conflict of interest.

Streptococcus pneumoniae is an important human pathogen causing infections of the local mucosa, like otitis media and sinusitis, and even more severe diseases like community-acquired pneumonia (CAP) and meningitis (1, 2). Yearly, 2 million people

in the United States suffer from pneumococcal infections, resulting in $4 billion in costs (3). It is estimated that 900,000 of these infections are caused by drug-resistant strains (3). The Centers for Disease Control and Prevention considers *S. pneumoniae* a serious threat (3). In addition to resistance, bacteria can also survive lethal antibiotic treatment by developing antibiotic tolerance (the ability to survive antibiotic treatments longer), for example, due to a lower killing rate without a change in the MIC (4, 5). Despite being frequently overlooked, antibiotic tolerance can set the stage for the development of genetic resistance and is associated with therapy failure (5–7). Regardless of the numerous reports of persistence in a variety of bacterial species (8–11), very little is known about antibiotic tolerance, and, to the best of our knowledge, nothing is known about antibiotic tolerance through persistence, in *S. pneumoniae* (12–16).

Persistent bacteria are a subpopulation of cells that transiently switch to a nongrowing state that enables them to survive treatment with a bactericidal drug concentration. Persisters are phenotypic variants within the bulk population but are genetically identical (4, 17, 18). As a consequence, persisters can transform back into antibiotic-susceptible bacteria and, after the antibiotic pressure is removed, reconstitute a population that displays an antibiotic tolerance identical to that of the starting culture (4, 18). Persister cells seem to be a universal feature of clonal life forms. Not only are they identified in many, if not all, bacterial species that have been studied, but also eukaryotic cancer cell lines and yeast populations contain drug-tolerant phenotypic variants (18–20). Many studies indicate the clinical relevance of persistence (7, 17, 18, 21, 22). Bartell et al. observed a link between high persister variants of *Pseudomonas aeruginosa*, long-term establishment of *P. aeruginosa* in the cystic fibrosis lung environment, and treatment failure (23). Van den Bergh et al. demonstrated the role of metabolic homeostasis, and more specifically of the respiratory complex I, as an important promoter of antibiotic persistence *in vitro* (24). Similarly, Fuentes et al. indicated the importance of the respiratory and fermentative metabolism in the preservation of heterogeneity within bacterial cultures and in adaptation of bacteria to environmental changes (25). Another important clinical consequence is that persistence is a driver toward the development of antibiotic resistance (18). Clearly, persisters constitute a viable pool that facilitates resistance development by prolonging the presence of viable bacteria during antibiotic treatment (26, 27), but various other mechanisms have been suggested (6, 8, 10, 26, 28). For example, Windels et al. and Huo et al. identified that the increased mutation rate in highly persistent strains promotes evolution toward antibiotic resistance (6, 29), and Levin-Reisman et al. indicated the role of epistasis between antibiotic persistence and resistance mutations (28).

The presence of antibiotic-tolerant persisters in *S. pneumoniae* has not been investigated to date. In part, the lack of understanding of persistence in *S. pneumoniae* stems from the self-limiting nature of this bacterium *in vitro* (30). Two suggested causes of the fast decrease in survival after entering the stationary phase are the enzymes pyruvate oxidase (SpxB) and autolysin (LytA) (31, 32). Pyruvate oxidase is the major producer of $H_2O_2$ as a by-product of the aerobic metabolism of *S. pneumoniae*, but *S. pneumoniae* lacks the neutralizing enzyme catalase, which leads to *in vitro* death through an accumulation of $H_2O_2$ (31, 33–35). Autolysin, a cell wall-bound amidase that breaks down peptidoglycan, induces *in vitro* autolysis in stationary-phase cultures (32, 36–38). Antibiotic-tolerant persisters are mostly connected with recurrent and chronic infections, and the role of persisters in acute infections is not clear (7, 18). Most infections caused by *S. pneumoniae* have an acute nature. Nonetheless, *S. pneumoniae* is also, albeit to a lesser extent, the causative agent of chronic diseases, like chronic endobronchial infections in children (39–41), and it can reside in biofilms in the middle ear in children, causing recurrent and chronic otitis media (42–45). The role of persister cells, in both acute and chronic pneumococcal infections, needs to be elucidated in order to gain a better understanding of how *S. pneumoniae* evades elimination by antibiotic treatment (7, 18, 46).

In this study, we made a broad inquiry on the presence and behavior of persister cells in populations of diverse *S. pneumoniae* isolates. We succeeded in obtaining stable long-living *in vitro* cultures using specific growth conditions which allowed us to

set up prolonged antibiotic-induced killing studies without confounding the results with the self-limiting nature of *S. pneumoniae*. Using these killing studies together with heritability assays, the gold standard assays to determine persistence (4, 18), we proved the presence of high numbers of persister cells in reference strain D39 cultures. Lastly, we detected surviving cells after antibiotic treatment, assuming the presence of persister cells, in a variety of *S. pneumoniae* strains, including clinical isolates (CIs), demonstrating that persistence is widely present and highly variable in *S. pneumoniae*. Our study is the beginning for persistence studies in *S. pneumoniae* and will lead to better insights on the role of persistence in acute, chronic, and recurrent *S. pneumoniae* infections. In turn, a better understanding of the escape mechanisms of *S. pneumoniae* will finally lead to improved therapeutic options.

## RESULTS

**Specific growth conditions allow long-living *in vitro* cultures of *S. pneumoniae*.** To study persistence, prolonged *in vitro* antibiotic-induced killing studies are required. Especially when examining antibiotic survival in stationary phase, long-living cultures are needed and any confounding effects of mortality through the self-limiting nature of *S. pneumoniae* must be avoided. In order to prevent *in vitro* death in the absence of antibiotics and to obtain a stable bacterial culture, we followed two routes targeting the suggested effectors of self-limitation in *S. pneumoniae*. First, we added catalase to neutralize the produced $H_2O_2$, we constructed an *spxB* knockout mutant to inhibit the expression of pyruvate oxidase, or we applied hypoxic incubation (5% $CO_2$–0.1% $O_2$–94.9% $N_2$) to inhibit the pneumococcal aerobic metabolism and thus the production of $H_2O_2$ by pyruvate oxidase (31, 33). Second, we used choline chloride supplementation to prevent the binding of autolysin to the cell wall or we used a *lytA* knockout mutant to inhibit the expression of autolysin (32, 36–38).

Despite the common knowledge and the regular growth in exponential phase (30, 47), a phase of significant killing after 8 to 48 h of incubation was unavoidable in the commonly used growth media brain heart infusion broth (BHI) and Todd-Hewitt broth supplemented with 0.5% yeast extract (THY), with or without applying the different strategies to counteract pyruvate oxidase or autolysin (see Fig. S1 in the supplemental material). While a section of the stationary phase was reasonably stable (5- to 20-fold reduction in bacterial concentration from 16 to 24 h) when adding catalase or using a *lytA* knockout mutant, the bacterial concentration was nevertheless reduced up to 1,000-fold before reaching such a stable period. Surprisingly, when using a less common growth medium, Mueller-Hinton broth supplemented with 5% lysed horse blood (MHL), *in vitro* self-limitation was mostly absent during 32 h of incubation (Fig. 1). Only after 32 h did a strong death phase occur, with a 10,000- to 1,000,000-fold reduction in bacterial viability. Also, counteracting pyruvate oxidase and autolysin by applying the proposed strategies did not significantly impact survival (Fig. S1). Therefore, MHL seems to be the optimal liquid growth medium to obtain a stable long-term bacterial culture.

Culture conditions are important for *S. pneumoniae* survival *in vitro*, and the use of the liquid growth medium MHL results in stable survival. Therefore, we optimized a model based on MHL as growth medium (Fig. 2). To validate the model for survival over 24 h, growth curves were obtained for five *S. pneumoniae* strains (Fig. 3). Strains were either commonly used lab strains (TIGR4, ATCC 49619, and R6) or lab strains from previous studies (85 and 88) (48). While these strains show small differences in lag phase, growth rate, and maximal bacterial concentration, overall survival over 24 h is stable among all strains (Fig. 3). In conclusion, the optimized long-living model results in a stable bacterial culture until 24 h of growth, which enabled us to execute prolonged time-killing experiments without the self-limiting nature of *S. pneumoniae* as a confounding factor. We applied this model in all further experiments described in this article.

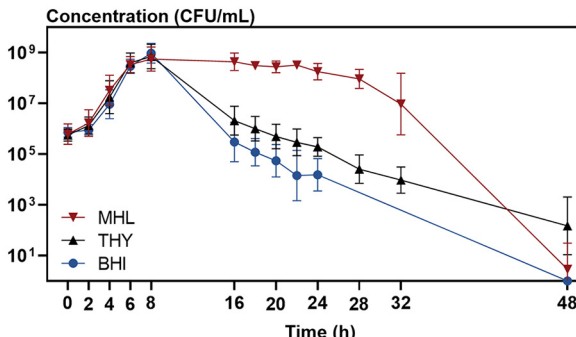

**FIG 1** MHL abolishes the self-limiting *in vitro* nature of *S. pneumoniae*. We compared the planktonic growth curves of *S. pneumoniae* in brain heart infusion broth (BHI), Todd-Hewitt broth supplemented with 0.5% yeast extract (THY), and Mueller-Hinton broth supplemented with 5% lysed horse blood (MHL). Survival was higher when *S. pneumoniae* was grown in MHL than in THY or BHI. The experiments were performed in triplicates, and values are presented as means ± standard deviations ($n = 3$).

**Persisters are widely present in *S. pneumoniae* reference strain D39 cultures and highly dependent on growth phase and type of antibiotic.** To study persister cells, the concentration of the applied bactericidal antibiotics needs to be well above the MIC to invoke killing of sensitive cells. Reference strain D39 is sensitive to amoxicillin, cefuroxime, moxifloxacin, and vancomycin—clinically relevant antibiotics of various classes—according to the EUCAST breakpoints (Table S1). To evaluate antibiotic-induced killing, applying an excess of such MICs is thus straightforward in further experiments (49–52). Along with the selection of antibiotics and their concentrations, we tested different growth phases for treatment of *S. pneumoniae* with the antibiotics. Three growth conditions are frequently used to score persister levels: the stationary phase, the diluted stationary phase, and the exponential phase (53–55). The protocol

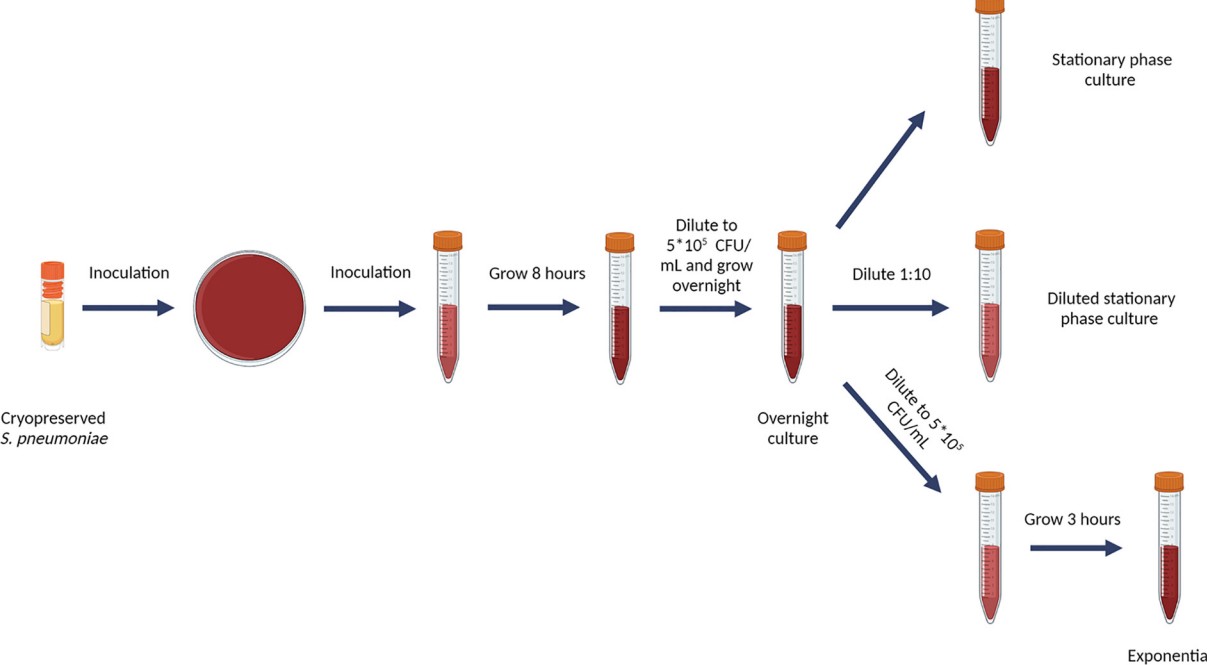

**FIG 2** Schematic overview of the long-living *in vitro* culturing protocol that results in stable bacterial cultures. Cryopreserved *S. pneumoniae* bacteria are plated on a blood agar plate, followed by inoculation in a tube with MHL. After 8 h of static incubation, the culture is diluted to $5 \times 10^5$ CFU/mL and grown overnight. The overnight culture either is directly used as a stationary-phase culture, diluted 1:10 in fresh MHL to act as a diluted stationary-phase culture, or is diluted to $5 \times 10^5$ CFU/mL in fresh MHL and grown for 3 h to obtain an exponential-phase culture.

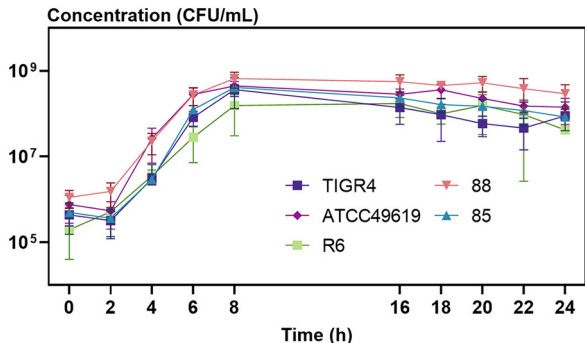

**FIG 3** Various *S. pneumoniae* lab strains show robust growth dynamics for up to 24 h of incubation using the optimized long-living *in vitro* model. Planktonic growth curves of *S. pneumoniae* TIGR4, ATCC 49619, R6, 85, and 88 as a function of time show small differences in lag phase, growth rate, and maximal bacterial concentration but a stable survival over 24 h. The experiments were performed in triplicates, and values are presented as means ± standard deviations ($n = 3$).

we used to obtain these different growth phases is described in Fig. 2. Briefly, cryopreserved *S. pneumoniae* bacteria were plated on a blood agar plate, followed by inoculation into a tube with MHL. After 8 h of static incubation, the culture was diluted to $5 \times 10^5$ CFU/mL and grown overnight. The overnight culture was either directly used as a stationary-phase culture, diluted 1:10 in fresh MHL to act as a diluted stationary-phase sample or was diluted to $5 \times 10^5$ CFU/mL in fresh MHL and grown 3 h to obtain an exponential-phase culture.

To evaluate the minimal dose needed to kill sensitive *S. pneumoniae* D39 cells within 5 h, we obtained dose-dependent kill curves by treating *S. pneumoniae* with increasing antibiotic concentrations, i.e., 5-, 10-, 20-, 100-, and 200-fold the MIC. Stationary-phase cultures proved insensitive to any of the antibiotics used, even at the highest dose, but treatment of diluted stationary-phase samples resulted in significant killing of sensitive cells at or above a concentration of 5-fold the MIC (Fig. 4). The independence on antibiotic concentration, once a sufficient dose is reached to kill sensitive cells, is a typical observation indicating a role for persistence, while strong correlations would point toward antibiotic resistance as the underlying cause of survival (4). In this case, the independence on antibiotic concentration could be the first indication of the presence of persister cells within *S. pneumoniae* D39 cultures. For the remainder of our

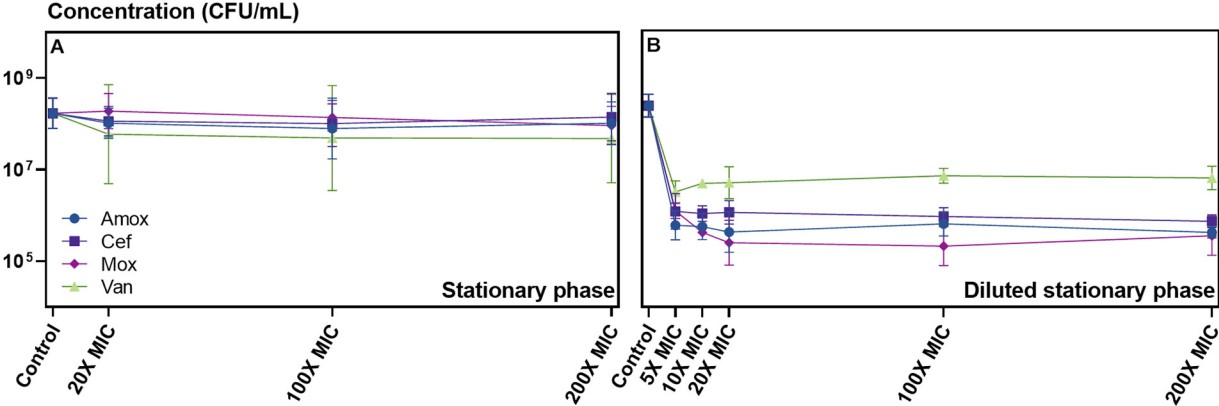

**FIG 4** *S. pneumoniae* D39 in stationary phase is insensitive to antibiotic treatment, while a dose of 5-fold the MIC is sufficient to kill sensitive cells in a diluted stationary-phase culture. Shown are dose-dependent kill curves with amoxicillin, cefuroxime, moxifloxacin, and vancomycin of planktonic *S. pneumoniae* D39 in stationary-phase (A) or diluted stationary-phase (B) samples. Antibiotic treatments lasted for 5 h before survivors were enumerated. Applied concentrations were 5-, 10-, 20-, 100-, and 200-fold the MIC (respectively, 0.03, 0.07, 0.14, 0.70, and 1.40 µg/mL for amoxicillin; 0.14, 0.27, 0.54, 2.7, and 5.4 µg/mL for cefuroxime; 1.4, 2.9, 5.8, 29, and 58 µg/mL for moxifloxacin; and 2.2, 4.4, 8.8, 44, and 88 µg/mL for vancomycin). The *y* axis of panel B corresponds to the *y* axis of panel A. The experiments were performed in triplicates, and values are presented as means ± standard deviations ($n = 3$).

work, we applied concentrations of 100-fold the MIC to ensure proper killing of sensitive cells and because a lower antibiotic concentration could lead to slower killing of normal cells, which would result in longer treatment times needed to reach the persister plateau in function of time.

**Streptococcus pneumoniae D39 cultures contain high numbers of persisters.** To investigate the presence of persister cells, we followed the survival of cells as a function of time during antibiotic treatment. The so-called time-kill curves should show a single rate of killing (uniphasic pattern) if the bacterial culture is fully susceptible to the antibiotic and lacks any subpopulation with increased tolerance (persister cells). However, if a subpopulation of antibiotic-tolerant persister cells is present within the susceptible population, we expect distinctly different killing rates to be apparent in the time-kill curves (biphasic pattern).

As stationary-phase cultures did not show any killing (Fig. 4), we performed these time-kill assays on diluted stationary-phase and exponentially growing samples. Upon dilution of stationary-phase cultures, antibiotic treatment killed 90 to 99.99% of the cells of strain D39 over an 8-h period, depending on the antibiotic (Fig. 5). We observed similar killing after an 8-h treatment of exponentially growing *S. pneumoniae* D39, but when treatment was prolonged to 24 h, antibiotic treatment killed an additional 3 orders of magnitude of the exponentially growing cells (Fig. 5). Mathematical analyses of the entire data set, with a global model containing a condition-dependent structure, showed that the biphasic killing model is superior to the uniphasic model in describing the data (analysis of variance [ANOVA; F test], $P = 1.58e-84$ [Table S2]), which implies that the sensitive *S. pneumoniae* D39 population contained persister cells. When each condition (growth phase $\times$ antibiotic) was analyzed separately, the biphasic model was significantly preferred over the uniphasic model for describing the data from all conditions ($P \leq 0.05$), except for data from treatment with cefuroxime and vancomycin in the diluted stationary growth phase. While this might indicate that including a second killing rate does not improve the models for these conditions, $P$ values were close to significance ($P = 0.1641$ and $0.1074$, respectively) and various test statistics (Akaike information criterion [AIC], Bayesian information criterion [BIC], and log likelihood [LogLik]) were either inconclusive or in favor of the biphasic model (Table S2). Overall, we detected

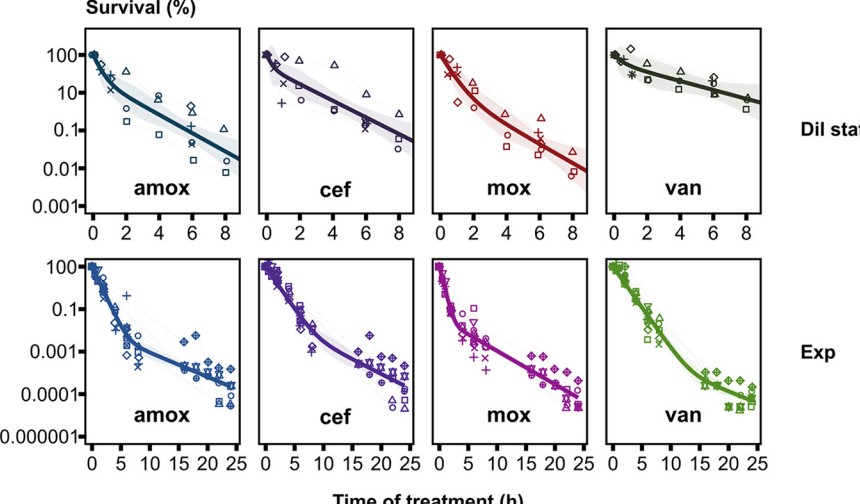

**FIG 5** Biphasic killing pattern upon antibiotic treatment indicates presence of persister subpopulations in *S. pneumoniae* D39 cultures. Fitting of a nonlinear fixed-effect model to log-transformed survival data upon treatment with amoxicillin, cefuroxime, moxifloxacin, and vancomycin against *S. pneumoniae* D39. Diluted (1:10) stationary-phase (Dil stat) and exponential-phase (Exp) bacteria were treated for 8 or 24 h with the antibiotic (100-fold the MIC; 0.70 $\mu$g/mL for amoxicillin, 2.7 $\mu$g/mL for cefuroxime, 29 $\mu$g/mL for moxifloxacin, and 44 $\mu$g/mL for vancomycin). Symbols show the individual repeats (time point connected and in the same shape if coming from the same repeat), and bold lines show the fitted biphasic killing curves $\pm$ 95% confidence intervals (shades) ($n \geq 3$).

relatively high persister levels, when the biphasic model was preferred over the uniphasic model, ranging from 13.74 to 24.31% for amoxicillin and moxifloxacin in the diluted stationary growth phase, compared to lower levels (ranging from 0.02 to 0.5%) in the exponential growth phase. The killing rates of persister cells (0.25 to 0.58 $h^{-1}$) were comparable between the different conditions and were 3- to 8-fold lower than the killing rates of normal cells (0.89 to 3.78 $h^{-1}$) (Table S3).

**The antibiotic-tolerant *S. pneumoniae* persisters were transient and nonheritable.** While biphasic killing patterns are the gold standard to identify persistence, theoretically, such surviving cells could still be the result of emerging resistance or of mutants that display an increased population-wide tolerance. To confirm the presence of persisters, we performed so-called heritability assays (4). We retested some of the surviving clones of the initial time-kill assay in a subsequent round of antibiotic treatment. If *S. pneumoniae* were resistant, the MIC value would have been increased, and if the persister phenotype were inherited and passed to the entire population of daughter cells, an increased survival would have been observed during the subsequent killing assays. During these subsequent killing assays, we observed a similar survival of randomly selected clones that survived the initial killing assay (i.e., supposed persisters) (Fig. 6), a similar killing dynamic pattern (Fig. S2), and MIC values that remained unchanged (Table S1) compared to the original culture. Thus, the surviving cells that we observed were genuine persister cells showing only a transient antibiotic tolerance, as regrown cultures show characteristics similar to those of the culture of origin.

**Antibiotic-tolerant cells are both prevalent and highly variable among *S. pneumoniae* clinical isolates.** Having established the presence of persisters in strain D39 in an optimized setup, we wondered whether other *S. pneumoniae* strains could survive antibiotic treatment, presumably through the presence of persister cells, and how this phenotype varies within the *S. pneumoniae* species. To answer these questions, we obtained a set of 10 clinical isolates (CIs 1 to 10) representing 10 different serotypes with known origin in addition to the already available strains (D39, TIGR4, ATCC 49619, R6, 85 [48], and 88 [48]). Clinical isolates were isolated either from patients suffering

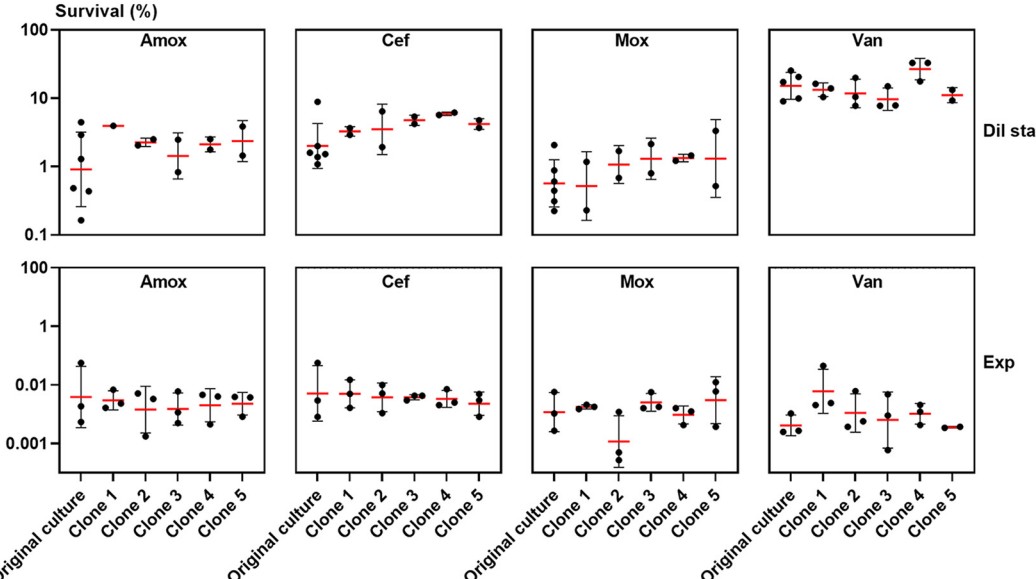

**FIG 6** The antibiotic tolerance of surviving *S. pneumoniae* cells is transient and nondeterministically inherited by daughter cells. Antibiotic-tolerant *S. pneumoniae* D39 clones were recovered after 6 (Dil stat) or 18 (Exp) hours of treatment during the initial time-kill assay, regrown without antibiotics, and preserved at −80°C. For these clones arising from potential persister cells, survival was determined after 6 (Dil stat) or 18 (Exp) hours of antibiotic treatment with amoxicillin, cefuroxime, moxifloxacin, and vancomycin in the diluted stationary or the exponential growth phase. Survival of the randomly selected clones was similar to that of the original culture (mixed-effect analysis; clone 1 was excluded from the analysis for amox—Dil stat because we had only one data point). The experiments were performed in duplicates or triplicates, and values are presented as means ± standard deviations (*n* ≥ 2).

**TABLE 1** *S. pneumoniae* strains used during this study[a]

| Strain | Serotype | Origin | Clinical diagnosis | Comorbidity | Specimen | Hospitalization status |
|---|---|---|---|---|---|---|
| TIGR4 | 4 | ATCC BAA-334 | | | | |
| D39 | 2 | NCTC 7466 | | | | |
| ATCC 49619 | 19F | ATCC 49619 | | | | |
| R6 | 2- | NCTC 13276 | | | | |
| 85 | 14 | Cools et al. (48) | | | | |
| 88 | 5 | Cools et al. (48) | | | | |
| 1 | 19F | Sciensano | Carriage | Cystic fibrosis | NPH | AMB |
| 2 | 11A | Sciensano | Carriage | Immunodeficient | NAS | HOS |
| 3 | 23B | Sciensano | Carriage | Cystic fibrosis | NPH | AMB |
| 4 | 19A | Sciensano | Conjunctivitis | NA | EYE | AMB |
| 5 | 6C | Sciensano | CAP | | BRA | HOS |
| 6 | 3 | Sciensano | COPD exacerbation | COPD | SPU | HOS |
| 7 | 23A | Sciensano | Sinusitis | | SIN | HOS |
| 8 | 9N | Sciensano | CAP | Chronic alcoholism | SPU | HOS |
| 9 | 16F | Sciensano | COPD exacerbation | COPD | SPU | AMB |
| 10 | 35B | Sciensano | Carriage | | SPU | HOS |

[a]NA, not available; CAP, community-acquired pneumoniae; COPD, chronic obstructive pulmonary disease; NPH, nasopharyngeal aspirate/swab; NAS, nasal swab; BRA, endotracheal/bronchial aspiration; SPU, sputum; SIN, sinus; AMB, ambulatory; HOS, hospitalized.

pneumococcal disease (CIs 4 to 9) or from patients that were only carriers (CIs 1 to 3 and CI 10). The infections that were caused by these strains were conjunctivitis (CI 4), community-acquired pneumonia (CAP) (CI 5 and CI 8), and sinusitis (CI 7), or the CI was isolated from patients during a chronic obstructive pulmonary disease (COPD) exacerbation (CI 6 and CI 9). Three out of 4 patients that carried *S. pneumoniae* suffered from comorbidities like cystic fibrosis or immunodeficiency, and 3 out of 6 patients that suffered from pneumococcal disease had comorbidities like COPD or chronic alcoholism (Table 1).

All strains were susceptible to amoxicillin, cefuroxime, moxifloxacin, and vancomycin, except for strain 85, which displayed cefuroxime resistance (MIC = 5.215 $\mu$g/mL), and CI 7, which displayed a MIC for moxifloxacin just above the resistance threshold (MIC = 0.637 $\mu$g/mL) according to the EUCAST breakpoints (Table S4). To screen for survival, *S. pneumoniae* strains were challenged with either amoxicillin or vancomycin (100-fold the MIC) for 8 h in the diluted stationary or exponential growth phase in order to determine survival (Fig. 7). Overall, we observed higher survival fractions for *S. pneumoniae* strains treated in the diluted stationary phase than for those treated in the exponential growth phase. Survival after treatment with vancomycin was higher than after challenging *S. pneumoniae* with amoxicillin in the diluted stationary phase, but survival fractions were comparable after treatment of exponentially growing bacteria. When we compared the survival of *S. pneumoniae* within a condition (growth phase $\times$ antibiotic), we detected large variations between strains ranging over 4 orders of magnitude for each condition. Strain R6, an unencapsulated reference strain, displayed the overall highest survival, while CI 10, an isolate from carriage, displayed the lowest. For all conditions, survival was significantly higher for CI 1 than for CI 10; both were carriage isolates (one-way ANOVA, $P \leq 0.01$). No significant differences were detected, under any condition, between the isolates from acute infections ($P > 0.05$), except for CI 4, which had a significantly lower survival than CI 7 after treatment with vancomycin in the diluted stationary phase ($P = 0.0461$). After the determination of survival levels of clinical isolates in an optimized setup, we assumed that the large differences in survival levels reflected differences in persistence among the tested *S. pneumoniae* strains. Despite the indications from the optimization with reference strain D39 that surviving cells were persisters, it remains possible that killing dynamics were strongly different in some of the clinical strains so that survival levels at one time point do not correctly reflect differences in persistence levels. For example, a difference in killing rates of normal cells and persisters can result in similar survival levels after antibiotic treatment, but with no or a different level of persisters. Since our current work could not test this more complex explanation, we therefore

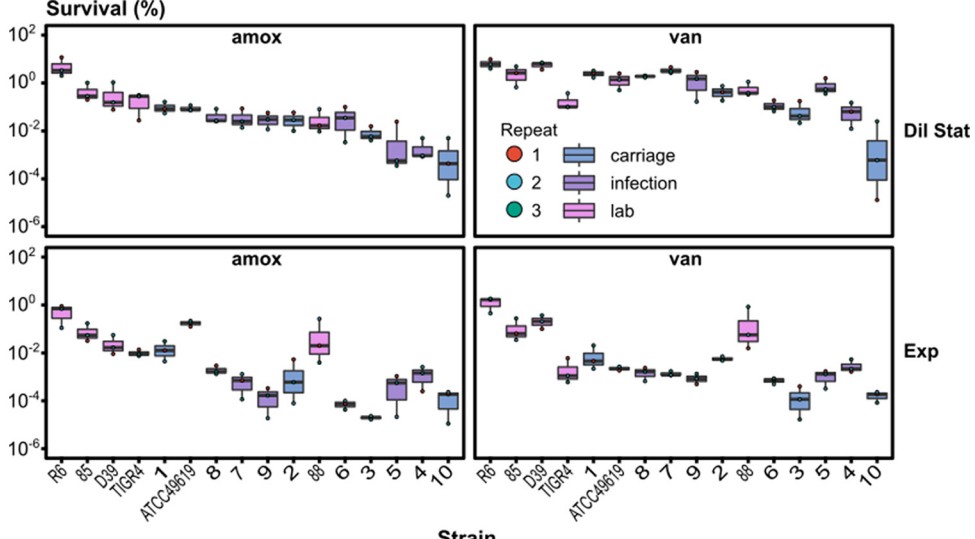

**FIG 7** Antibiotic-tolerant cells are widely present after antibiotic treatment and highly variable among *S. pneumoniae* strains from different sources. Survival is highly variable in *S. pneumoniae* strains after challenge with 100-fold the MIC of amoxicillin or vancomycin. An 8-h antibiotic treatment was started after a 1:10 dilution of stationary-phase bacteria or after dilution of stationary-phase bacteria to $5 \times 10^5$ CFU/mL followed by 3 h of growth (exponentially growing bacteria). Starting concentrations were in the range of $5 \times 10^6$ to $8 \times 10^7$ CFU/mL for the diluted stationary phase and in the range of $2 \times 10^5$ to $1 \times 10^7$ CFU/mL for the exponential cultures. Strains are ordered based on their survival after treatment with amoxicillin in the diluted stationary phase. Log-transformed data are shown as a boxplot of the mean $\pm$ standard deviation for each strain. The experiments were performed in triplicates, and each individual measurement is given as a dot according to the repeat ($n = 3$).

conclude that survival, likely through persistence, is highly variably in *S. pneumoniae* strains from different sources.

Given the strong variation between strains, we wondered whether some strains show antibiotic survival specific to one condition or whether survival levels of these strains can be correlated between different conditions. Survival levels correlated strongly between the growth phases (diluted stationary and exponential) and the antibiotics (amoxicillin and vancomycin) (Fig. 8 and Fig. S3). Surprisingly, we also detected a small positive correlation (Pearson correlation [$R^2$] ranging from 0.032 to 0.26; $P \le 0.05$ for all conditions, except for vancomycin in the diluted stationary phase)

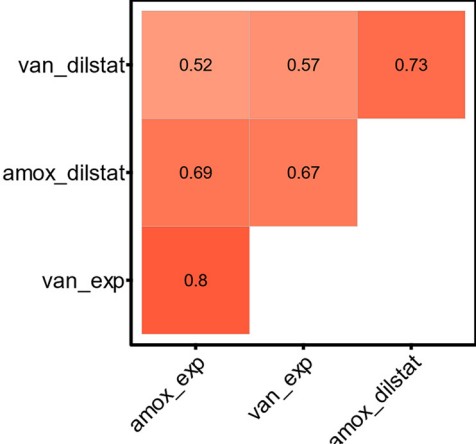

**FIG 8** Correlation analysis of survival between different conditions showed strong correlations between antibiotics (amoxicillin and vancomycin) and growth phases (diluted stationary and exponential growth phase). Shown is a correlation matrix between the survival fractions under 4 different conditions: treatment with amoxicillin or vancomycin in the diluted stationary (dilstat) or exponential (exp) growth phase. Pearson correlation coefficients (*R*) are given for each correlation.

between increased growth and increased survival, which does not substantiate the common belief that slow growth induces persister formation (56–62). Finally, we detected a small negative correlation between initial CFU before treatment and survival ($R^2$ ranging from $2.1 \times 10^{-6}$ to 0.11; $P < 0.05$ for amoxicillin in the exponential phase), implying that survival was higher when antibiotic treatment was started on a culture with a lower bacterial concentration (Fig. S3). After screening different *S. pneumoniae* strains for survival, we assume that persisters are widely present in *S. pneumoniae* cultures from different sources (lab strains and clinical isolates from infection and carriage), with strong variations between strains within a condition.

## DISCUSSION

Our study presents a broad characterization of persistence in *S. pneumoniae*. We confirmed the hypothesis that persister cells are present in *S. pneumoniae* cultures by finding strong indications of a biphasic killing pattern, the hallmark of persistence, after treatment with four clinically relevant antibiotics with different modes of action. Additionally, the surviving persisters in *S. pneumoniae* show no inheritable resistance, as tolerance to the antibiotics was not passed from the initial persister cell to the subsequent generation during the heritability assays and the MIC values remained unchanged. Finally, a set of clinical isolates was screened for survival after antibiotic treatment. Here, we identified large variations in survival levels among different strains, suggesting that differences in survival levels are a result of differences in persistence. After the optimization experiments with strain D39 and the screening of a variety of *S. pneumoniae* strains, we assume that persistence is a general trait in *S. pneumoniae* cultures.

Most insights in persistence have been gathered using Gram-negative bacteria, like *Escherichia coli*, *Salmonella enterica* serovar Typhimurium, and *Pseudomonas aeruginosa* (8, 10, 11, 18). However, different studies have indicated a role for persistence in Gram-positive bacteria as well (18, 63–65), more specifically, in various reports on streptococcal species (18, 66). Persisters in *Streptococcus mutans*, a cariogenic oral bacterium, are tolerant to a wide variety of antibiotics (67) but also to a dental caries defensive agent, dimethylaminododecyl methacrylate (DMADDM) (68), and to other antibacterial monomers used in dental medicine (69). As in nonstreptococcal bacterial species, toxin-antitoxin systems are involved in the formation of persister cells in *S. mutans*, as well as the quorum-sensing competence-stimulating peptide (CSP) (67, 70). Similarly, antibiotic-tolerant persisters were identified in the zoonotic pathogen *S. suis* by Willenborg et al. (52) and in the opportunistic human pathogen *S. faecalis* as early as 1979 by Soriano and Greenwood (71). Additionally, persistence in *S. pyogenes* was observed by Wood et al. in stationary-phase *in vitro* cultures (72), and Martini et al. detected persister cells in *S. pyogenes* biofilms treated with antimicrobials (73). Despite the various reports on persistence in other species of the *Streptococcus* genus, very little is known about antibiotic tolerance, and more specifically about persistence, in the Gram-positive bacterium *S. pneumoniae* (13–16).

*S. pneumoniae* is well known to cause acute infections, while antibiotic tolerance and persistence are mostly connected with recurrent and chronic infections (2, 7). Nonetheless, we expected to find persister cells, as persistence has been identified in many, if not all, bacterial species that have been studied and *S. pneumoniae* causes, to a lesser extent, also recurrent and chronic infections (40, 41, 44, 74). After mathematical analyses of the killing data of *S. pneumoniae* D39, we detected a biphasic killing pattern, which indicates the presence of persister cells. A major advantage of our approach is that we determined persister fractions, and killing rates, by mathematical analysis based on kill curves over a prolonged treatment period, which enabled us to take into account the killing pattern rather than determine the persister fraction based on a single time point. The characteristics of persistence differed between growth phases and antibiotics. Survival levels were vastly higher in diluted stationary-phase cultures than for exponentially growing bacteria for all examined strains, as we expected, because persistence is mostly linked to dormancy and bacteria from the diluted stationary phase recently came

out of the stationary phase and could therefore be less metabolically active (17, 42, 75). The difference in persistence between antibiotics can be attributed to the different modes of action of the antibiotics. Interestingly, treatment of diluted stationary-phase cultures with moxifloxacin, a fluoroquinolone that targets the DNA synthesis of bacteria and is less dependent on cell growth than β-lactams, resulted in the lowest level of persisters (13.74%) in diluted stationary-phase cultures (46, 76, 77). Overall, we observed higher persister fractions than previously described, but a potential explanation is the difference in approach: we used mathematical analysis based on prolonged time-kill curves rather than on survival at a single time point. If we determine the persister fraction based on survival fractions after a fixed period of treatment, for example, after 8 h, fractions approximate 0.01 to 1% as is described for *S. mutans* or *E. coli* stationary-phase cultures (54, 67). Altogether, we determined that persisters are widely present in *S. pneumoniae* reference strain D39 cultures and that *S. pneumoniae* persistence is highly dependent on growth phase and the type of antibiotic.

Finally, a set of *S. pneumoniae* strains was screened for survival after antibiotic treatment. In total, 16 strains were screened, including 6 lab strains (D39, TIGR4, ATCC 49619, R6, 85 [48], and 88 [48]) and 10 clinical isolates (CIs 1 to 10). This set of *S. pneumoniae* strains is diverse, with different serotypes and from different origins (lab, infection, or carriage). Survivors were detected for all strains in various levels, ranging from 0.001% to 10% surviving cells, assuming that persistence is widely present but also highly variable in *S. pneumoniae* cultures. Hofsteenge et al. and Stewart and Rosen detected comparable variations in survival after antibiotic treatment of a set of natural and environmental *E. coli* isolates, respectively (78, 79). Similarly, Barth et al. observed a high heterogeneity of persister cell formation among *Acinetobacter baumannii* isolates (80). We analyzed the survival data for correlations (Pearson) between the growth phases and antibiotics. The strong correlation between the growth phases indicates that these conditions affect survival in a similar way, potentially because they both induce growth and metabolic activity. Furthermore, survival data for both types of antibiotics, amoxicillin and vancomycin, correlate well, potentially as a consequence of the fact that they both target the bacterial cell wall synthesis. Surprisingly, we saw a slight positive correlation between how fast the strains grow and how well they survive antibiotic treatment, especially for exponentially growing bacteria. These correlations imply that if the bacteria grew faster, they survived antibiotic treatment better. This was unexpected, as persistence is linked to dormancy of bacterial cells and we expected that if cells were less actively dividing and less metabolically active, they would survive antibiotic treatment better. However, different studies state that global metabolic dormancy is not solely responsible for tolerance (56–59, 81). For example, Stapels et al. and Peyrusson et al. demonstrated the presence of nondividing but metabolically active *Salmonella* and *Staphylococcus aureus* persisters, respectively, during intracellular infections (60, 61), and Goneau et al. stated that antibiotic tolerance is caused more likely by selective target inactivation than by global metabolic dormancy in uropathogens (62). With the screening of a variety of *S. pneumoniae* strains, we assume that persistence is widespread and diverse among *S. pneumoniae* cultures. Further screening of clinical isolates is necessary to be able to correlate persistence with serotype and origin, but also to study the role of *S. pneumoniae* persisters in treatment outcome.

With this study, we made a broad characterization of persistence in *S. pneumoniae*. First, we obtained long-living *in vitro* cultures of *S. pneumoniae*, eliminating its self-limiting nature. Second, we detected the presence of a biphasic killing pattern after analyzing antibiotic-induced time-kill assays, the hallmark of persistence, and we proved that *S. pneumoniae* persistence is transient and not heritable. Finally, we observed surviving cells, presumably persister cells, in *S. pneumoniae* strains from different origins, which assumes that persistence is widely distributed and highly variable among *S. pneumoniae* isolates. Our work advocates for higher interest in persistence in *S. pneumoniae* as a contributing factor for therapy failure and resistance development. Future studies should

gain better insights in the mechanisms of persister formation and improve knowledge about the clinical relevance of pneumococcal persisters. Therefore, further screening of clinical isolates and *in vivo* studies are required. The ultimate goal is to gain better insights into the role of persistence in acute, recurrent, and chronic *S. pneumoniae* infections, which will hopefully lead to improved therapeutic options.

## MATERIALS AND METHODS

**Bacterial strains and growth conditions.** *S. pneumoniae* strains used are listed in Table 1. *S. pneumoniae* was cultured statically in brain heart infusion broth (BHI; Neogen), Todd-Hewitt broth (BD Biosciences) supplemented with 0.5% yeast extract (THY; Gibco), or cation-adjusted Mueller-Hinton broth (Fluka) supplemented with 5% lysed horse blood (MHL; Oxoid) or on blood agar (BA) plates (tryptic soy agar [Neogen] supplemented with 5% defibrinated sheep blood [Oxoid]) at 37°C in 5% $CO_2$. Catalase (30,000 U/mL; MP Biomedicals) or choline chloride (Sigma-Aldrich) was added when specified. When specified, bacteria were grown under hypoxic conditions (5% $CO_2$–0.1% $O_2$–94.9% $N_2$) in a Whitley H35 Hypoxystation. *Escherichia coli* strain DH5$\alpha$ was cultured under shaking in Luria-Bertani broth (Lennox) (LB; Sigma-Aldrich) at 37°C and 175 rpm.

**Planktonic growth and enumeration of bacteria.** Bacteria were grown in different media with or without catalase (1,000 U/mL) or choline chloride (10 mM) supplementation. At different time points, samples were taken and the bacterial concentration was determined according to the viable plate count (VPC) method. Briefly, a 1:10 serial dilution ($10^0$ to $10^{-6}$) was made in phosphate-buffered saline (PBS) in a 96-well plate. Three drops of 10 $\mu$L of a selection of dilutions was plated on BA and incubated for minimum of 24 h before colonies were counted and suspensions were enumerated.

**Long-living *in vitro* culturing.** Bacteria from cryopreservation were plated on a blood agar plate and incubated for 24 to 72 h, followed by subculturing in a tube with MHL for 8 h with a final concentration of 1 × $10^8$ CFU/mL. Then, bacteria were diluted to 5 × $10^5$ CFU/mL in fresh MHL and brought into the desired growth state. Stationary-phase bacteria were obtained by overnight growth (16 h). Diluted stationary-phase bacteria were obtained by overnight growth (16 h) and 1:10 dilution in fresh MHL. Exponential-phase bacteria were obtained by overnight growth (16 h), dilution to 5 × $10^5$ CFU/mL in fresh MHL, and 3 h of growth (Fig. 2).

**Construction of knockout mutants. (i) Vector construction.** The first and last 500-bp regions of the gene (*lytA* or *spxB*) were amplified from *S. pneumoniae* D39 chromosomal DNA by PCR using Q5 high-fidelity DNA polymerase (New England Biolabs). The kanamycin resistance cassette was amplified from pSt-K and the streptomycin resistance cassette from pGMC5-SM-RFP-PFurA-GFP-streptomycin. The PCR primers contained overhang sequences with the antibiotic resistance marker (kanamycin resistance cassette for *lytA* and streptomycin resistance cassette for *spxB*) and the pGEM-T Easy vector (Promega) (Table S5). The first and last 500 bp of the gene and the antibiotic resistance cassette were then introduced into the pGEM-T Easy vector using HiFi DNA assembly (New England Biolabs), resulting in plasmids pLytA and pSpxB (Fig. S4) and the plasmids were used to transform chemocompetent *E. coli* DH5$\alpha$. The resultant plasmid was verified by PCR and sequencing and used to transform *S. pneumoniae* D39.

**(ii) Transformation.** Precompetent *S. pneumoniae* cells were obtained by growing *S. pneumoniae* in THY to 3 × $10^8$ CFU/mL from a starting concentration of 1 × $10^6$ CFU/mL. Then, the bacterial suspension was diluted 1:100 in competence medium (THY supplemented with 0.2% bovine serum albumin and 0.01% $CaCl_2$), 10% glycerol (Sigma-Aldrich) was added, and bacteria were stored at −80°C. For transformation, precompetent *S. pneumoniae* was thawed, competence-stimulating peptide 1 (CSP-1) was added (2.5 $\mu$g/mL), and competence was induced by incubation at 37°C in a water bath. After 20 min, 200 ng of plasmid DNA was added and bacteria were incubated for an additional 60 min at 30°C and transferred to 37°C for 90 min before plating on BA containing 400 $\mu$g/mL of kanamycin (*lytA*) (Sigma-Aldrich) or 200 $\mu$g/mL of streptomycin (*spxB*) (Sigma-Aldrich). Resistant colonies were selected and the mutation was confirmed by sequencing. Knockdown of *lytA* and *spxB* was confirmed by quantitative PCR (qPCR) (Fig. S5).

**Antibiotic susceptibility.** The MICs of standard antibiotics were determined using a resazurin assay as described previously (48). Antibiotics used were amoxicillin (Sigma-Aldrich; beta-lactam antibiotic), cefuroxime (Sigma-Aldrich; beta-lactam antibiotic), moxifloxacin (Sigma-Aldrich; fluoroquinolone), and vancomycin (Sigma-Aldrich; glycopeptide), four clinically relevant antibiotics that are either commonly used to treat *S. pneumoniae* infections (amoxicillin, cefuroxime, and moxifloxacin) or used as a last resort, mostly in a hospital environment (vancomycin) (82–84). Briefly, a 1:2 serial dilution of the antibiotic was made in triplicates in MHL in a 96-well plate with a final volume of 100 $\mu$L. Then, 100 $\mu$L of a bacterial suspension was added to each well, except negative-control wells, to a final concentration of 5 × $10^5$ CFU/mL in 200 $\mu$L. Positive-control wells contained 200 $\mu$L of bacterial suspension (5 × $10^5$ CFU/mL) without antibiotics, and negative-control wells contained 200 $\mu$L of MHL without antibiotics or bacteria. Plates were incubated at 37°C and 5% $CO_2$ for 20 h before 20 $\mu$L of a 0.005% resazurin solution was added. Plates were further incubated for 90 min and fluorescence was measured ($\lambda_{em}$ = 590 nm; $\lambda_{ex}$ = 520 nm) using a spectrophotometer (Promega; Discover).

**Dose-dependent and time-kill curves.** To obtain dose-dependent kill curves, *S. pneumoniae* was treated for 5 h in the stationary or diluted stationary growth phase with five different antibiotic concentrations (5×, 10×, 20×, 100×, and 200× the MIC, respectively; 0.03, 0.07, 0.14, 0.70, and 1.40 $\mu$g/mL for amoxicillin; 0.14, 0.27, 0.54, 2.7, and 5.4 $\mu$g/mL for cefuroxime; 1.4, 2.9, 5.8, 29, and 58 $\mu$g/mL for moxifloxacin; and 2.2, 4.4, 8.8, 44, and 88 $\mu$g/mL for vancomycin). After 5 h, bacterial suspensions were centrifuged,

resuspended in PBS to wash away antibiotics, and enumerated by VPC. Colonies were counted after a minimum of 48 h of incubation. To obtain time-kill curves, *S. pneumoniae* was treated in the diluted stationary or exponential growth state with a fixed antibiotic concentration (100-fold the MIC). Bacterial suspensions were incubated for 8 or 24 h. At specified time points, bacterial suspensions were enumerated using VPC after centrifugation and resuspension in PBS to wash away the antibiotics. Colonies were counted after a minimum of 48 h of incubation.

**Heritability assay.** For each condition (growth phase × antibiotic), 5 clones from the initial time-kill experiment were isolated from the blood agar plate from the second killing phase (after 6 h of treatment for the diluted stationary phase and after 18 h of treatment for the exponential phase), regrown in fresh MHL without antibiotics, and stored at −80°C. These bacterial clones were subjected to the same protocol as in the initial time-kill assay. For one of these clones per condition, a time-kill curve was obtained and the MIC value was determined. For the other 4 clones, a fixed time point (6 or 18 h of treatment) was chosen to determine survival.

**Screening of lab strains and clinical isolates for survival.** A screening model was set up for clinical isolates based on the protocol used for strain D39. Briefly, *S. pneumoniae* was brought into the right growth phase (diluted stationary or exponential growth phase) followed by treatment with 100-fold the MIC of amoxicillin or vancomycin. After 8 h of antibiotic treatment, bacterial suspensions were enumerated using VPC after centrifugation and resuspension in PBS to wash away the antibiotics and survival fractions were calculated.

**Data analysis and statistics.** Student's *t* test, one-way ANOVA, mixed-effect analysis, or two-way ANOVA was used to compare continuous variables (MIC values, survival fractions, and time-kill curves) in GraphPad Prism version 9. A difference between two groups was considered statistically significant when the *P* value was $<0.05$. The R packages *nls.multstart*, *broom*, and *purrr* were used to analyze the time-kill curves mathematically by comparing two models of killing, a uniphasic model with a single killing rate and a biphasic model with two killing rates. The nonlinear fixed-effect model used the $\log_{10}$-transformed fraction of surviving cells. The biphasic model was based on the equation $\log(Y) = \log[(N - P_0)^{(-kn \times t)} + P_0^{(-kp \times t)}]$ and the monophasic model on $\log(Y) = \log[[(N)^{(-kn \times t)}]$ where $Y$ is survival fraction, $t$ is treatment time (in hours), $P_0$ is persister fraction at $t$ of 0, and $k_n$ and $k_p$ are the killing rates of normal and persister cells (per hour). Curves were considered biphasic if biphasic fitting was better than uniphasic fitting according to ANOVA (F test), the Akaike information criterion (AIC), the Bayesian information criterion (BIC), and the log likelihood (LogLik). The R package *ggcorrplot* was used to execute correlation analyses. The raw data are available on Zenodo via the following link: https://doi.org/10.5281/zenodo.7147832.

## SUPPLEMENTAL MATERIAL

Supplemental material is available online only.

**SUPPLEMENTAL FILE 1**, PDF file, 0.7 MB.

## ACKNOWLEDGMENTS

This research was funded by the Research Foundation-Flanders (FWO), grant numbers 12O1917N, V428917N, and 1513120N. B.V.d.B. and N.G. were funded by the FWO (postdoctoral fellowship 12O1922N and predoctoral fellowship 11E4721N, respectively).

We report no conflicts of interest.

The funders had no role in the design of the study; in the collection, analyses, or interpretation of data; in the writing of the manuscript, or in the decision to publish the results.

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
