## [Reviewer comments · Microbiology Spectrum]

Microbiology Spectrum

Antibiotic tolerance indicative for persistence is pervasive among clinical *Streptococcus pneumoniae* isolates and shows strong condition dependence

Nele Geerts, Linda De Vooght, Ioannis Passaris, Peter Delputte, Bram Van den Bergh, and Paul Cos

Corresponding Author(s): Paul Cos, University of Antwerp

Review Timeline:

Submission Date:	July 17, 2022
Editorial Decision:	August 11, 2022
Revision Received:	October 12, 2022
Accepted:	October 16, 2022

Editor: John Atack

Reviewer(s): The reviewers have opted to remain anonymous.

Transaction Report:

DOI: <https://doi.org/10.1128/spectrum.02701-22>

August 11, 2022

Prof. Paul Cos
University of Antwerp
Laboratory of Microbiology, Parasitology and Hygiene
Universiteitsplein 1
wilrijk, antwerpen 2610
Belgium

Re: Spectrum02701-22 (Antibiotic-tolerant persisters are pervasive among clinical *Streptococcus pneumoniae* isolates and show strong condition-dependence)

Dear Prof. Paul Cos:

Link Not Available

Sincerely,

John Atack

Journals Department
Reviewer comments:

Reviewer #1 (Comments for the Author):

Minor Comments

1. Lines 18-20- suggest to tone down language from nothing to little is known. Pretty much all known bacteria make persisters, what we have not really had is robust experimental means to examine persisters in pneumococcus, which this paper addresses.
2. Line 230- it is either resistant or not, cut minor. You could reword to say that is was just above the resistance threshold at MIC = X.

3. Figure 2- spelling errors in the figure - should be inoculation, Crypreserved should be cryopreserved.
4. Figure 4- I assume same y-axis for panels A and B but this should be made clear in figure legend or the figure.
5. Figure 6- all experiments supposedly have an n=3, but some appear to only have 1 or 2 graphed (i.e. amox clone 1 dilute stationary). If the datapoints overlap is it possible to have them side by side such that all datapoints can be visualized?
6. I think the y-axis in Figure 7 is incorrect or is not to scale. The scale goes from 1 to 0.01 to 0.001. Is the 1 supposed to be 0.1?

Major Comments

1. While of interest, the inclusion of the *lytA*/*spxB* detracts from the overall narrative. In my opinion, while these efforts were smart avenues to address the question at hand, the utilization of the alternative media that was used for the remainder of the manuscript is where the bulk of the experimental data is focuses. Cutting these sections would make the manuscript more concise and focused. To this end, I might revise figure 1 to indicate the Thy and MHL results. The *lytA*/*spxB* data could be placed in supplemental data. Figure 9 could also be moved to supplemental data.

2. I thought the data shown in Figure 7 to be very compelling as the kill kinetics really show some interesting differences. What these experiments lack is two (fairly minor) aspects. First, the number of bacteria from the 1:100 dilution (starting concentration) should be indicated here. Are they all from the same starting point or do they vary considerably in their starting CFU? Another aspect to consider would be to measure growth rate (doubling times) from these 1:100 dilutions of the strains. Are the strains that outgrow following dilution the slowest the most likely to survive? And vice versa, are the once that outgrow the fastest the more susceptible? Even if this does not correlate with the survival, this would be substantially strengthen the findings of this paper.

Reviewer #2 (Comments for the Author):

In the manuscript entitled, "Antibiotic-tolerant persisters are pervasive among clinical *Streptococcus pneumoniae* isolates and show strong condition-dependence", Geerts et al provide evidence for the presence of persister cells in *Streptococcus pneumoniae* D39 strain following antibiotic treatment. There were two supporting pieces of evidence - first, the bacterial population showed the characteristic biphasic killing pattern, and second, there was mostly no change in the MIC of persister cells relative to rest of the sensitive bacterial population. The number of persister cells varied depending on the growth phase of the bacteria, and the type of antibiotic used. While the authors attempt to demonstrate that this phenomenon is well conserved across different pneumococcal strains, they have not provided sufficient evidence to support this claim (more below). While this is an important study on a clinically relevant subject, the manuscript is missing some controls to convincingly report the persistence of persisters in *S. pneumoniae*. Additionally, the manuscript will be more robust if certain statements and overinterpretations are toned down.

Major Comments -

1. Lines 169 - 173:

- a. Authors state that significant killing of sensitive cells occurred at or above a concentration of 20X MIC. Did concentrations below that threshold not result in sufficient killing of sensitive cells in the observed time frame? Please provide data for the same. Further, it is unclear why a concentration of 100X the MIC was chosen for rest of the work - please elaborate on the rationale. For instance, why not 20X or 50X the MIC?
- b. A hallmark of persistence is that the size of the persister subpopulation is only weakly dependent on the antibiotic concentration. Is that true for this work? The authors have only showed proportion of persister cells formed at 100X the MIC, and not any other concentration.

2. Lines 195-197: The reported proportion of persister cells (13.74-60.08%) in dilute stationary phase cultures upon treatment with different antibiotics fall in a wide range, and perhaps needs another look. Since killing data for cefuroxime and vancomycin does not fit biphasic pattern better than uniphasic pattern, it is unclear whether the tolerant cells in these conditions are bonafide persisters. As such, it might be better to report this proportion only when the killing curves show a biphasic pattern i.e. upon treatment with amoxicillin and moxifloxacin.

3. Figure 6 and lines 208-212: It is unclear why the authors have used a different time point for comparing the proportion of cells that survive following antibiotic treatment when grown in exponential phase. It seems that for initial antibiotic treatment, cells were treated with antibiotic for 18 hours. But in the subsequent round, the corresponding time point was 6 hours. Why is that? Additionally, in Fig 6 - for diluted stationary phase cultures, most antibiotic treatments (except for vancomycin) have 2 data points each for persister clones. It is also unclear how ANOVA was performed with <3 data points.

4. Lines 216- 246 (and other places): The authors have shown that there is some survival of cells when additional strains and clinical isolates are treated with different antibiotics. However, they have not demonstrated that these surviving cells are persisters for the following reasons -
- Survival was reported for a time point as early as 3 hours following antibiotic treatment of exponentially growing bacteria. Even for D39 strain, at such an early time point (eg Fig 5) - the cells being killed are sensitive to antibiotics and not persisters. Thus, reporting these surviving cells as persisters (from data in Fig 7) seems misleading and inaccurate.
 - No data regarding the kinetics of antibiotic-induced killing has been shown. Is the killing pattern uniphasic or biphasic?
 - Is there any change in the MIC of the surviving cells? Is the survivability phenotype heritable? Without this data, no conclusions can be drawn on whether these surviving cells are persisters or not.
5. Lines 251-253, 320-325: A Pearson's correlation as small as 0.032 is perhaps more reflective of no correlation than positive correlation. Thus, in the absence of other supporting data to support positive correlation, the conclusion that the data "contradicts the common believe (belief) that slow growth induces persister formation" seems like an overinterpretation.
6. Lines 40-41, 331-333 etc: The assertion that a "broad range of genetic elements" are controlling persister generation is speculative. The potential conservation of the phenotype across diverse pneumococcal strains alone does not necessarily indicate that numerous genetic elements or accessory genes are responsible for persister formation. It is possible that a few near core genes are responsible for the phenotype.
7. Lines 292-293: Could the authors speculate on the reasons behind why persistence differed between growth phases and antibiotics.
8. (Optional): Is there any correlation between the resistance level of a strain (high vs low MIC) and its propensity to form persisters? For instance, strain 85 is resistance to cefuroxime and CI 7 is reported to be resistant to moxifloxacin. Are there any persisters present in these strains following treatment with these respective antibiotics? If so, what is the percent of the population that forms persisters?

Minor Comments -

- Lines 117, 121, and elsewhere. Nomenclature for gene name - the last letter should be italicized (spxB, lytA).
- Lines 90-91: While persister formation has not previously been investigated, there are some studies have investigated the antibiotic tolerance phenotype observed in pneumococcus. Since pneumococcus also causes chronic infections (as acknowledged later in the paragraph) and is a common colonizer, the statement that persisters may have been ignored because of the "acute" nature of pneumococcal infections seems misleading.
- Lines 362-363: Please report how much catalase and choline chloride were added.
- Fig 9 - It might be better to include this figure as a supplement figure as opposed to a main figure since it is tangential to the main findings being reported.
- Typos: Despite "being" frequently (line 55), set "the" stage (line 55), Another important clinical (line 74), round of "antibiotic" treatment (line 206), "specifically" about (line 283)

Staff Comments:

Preparing Revision Guidelines

Please return the manuscript within 60 days; if you cannot complete the modification within this time period, please contact me. If you do not wish to modify the manuscript and prefer to submit it to another journal, please notify me of your decision immediately so that the manuscript may be formally withdrawn from consideration by Microbiology Spectrum.

Reviewer #1 (Comments for the Author)

Minor Comments

1. Lines 18-20- suggest to tone down language from nothing to little is known. Pretty much all known bacteria make persisters, what we have not really had is robust experimental means to examine persisters in pneumococcus, which this paper addresses.

We agree with the reviewer that the sentence was a bit overstated so we adjusted the text.

Lines 19-21

In addition, persistence also contributes to the antibiotic crisis in many other pathogens, yet, in S. pneumoniae little is known about antibiotic-tolerant persisters and robust experimental means are lacking.

2. Line 230- it is either resistant or not, cut minor. You could reword to say that is was just above the resistance threshold at MIC = X.

We adjusted the text in accordance to your suggestion.

Lines 234-237

All strains were susceptible to amoxicillin, cefuroxime, moxifloxacin, and vancomycin, except for strain 85 which displayed cefuroxime resistance (MIC = 5.215 µg/mL) and CI 7 which displayed a MIC for moxifloxacin just above the resistance threshold (MIC = 0.637 µg/mL) according to the EUCAST breaking points (Table S4).

3. Figure 2- spelling errors in the figure - should be inoculation, Crypreserved should by cryopreserved.

We apologize for these errors and have adjusted all aforementioned comments in figure 2.

4. Figure 4- I assume same y-axis for panels A and B but this should be made clear in figure legend or the figure.

Indeed, the y-axes are depicted in a compact fashion as they are the same in both panels. We added this information to the legend of figure 4.

Figure legend 4

Figure 4: *S. pneumoniae* D39 in stationary phase is insensitive to antibiotic treatment while a dose of 5-fold the MIC is sufficient to kill sensitive cells in a diluted stationary phase culture.

Dose-dependent kill curves with amoxicillin (amox, blue dots), cefuroxime (cef, dark purple squares), moxifloxacin (mox, light purple diamonds) and vancomycin (van, green triangles) of planktonic S. pneumoniae D39 in (A) stationary phase or (B) diluted stationary phase samples. Antibiotic treatments lasted for five hours before enumerating survivors. Applied concentrations were 5-, 10-, 20-, 100- and 200-fold the MIC (respectively 0.03; 0.07; 0.14; 0.70 and 1.40 µg/mL for amoxicillin, 0.14; 0.27; 0.54; 2.7 and 5.4 µg/mL for cefuroxime, 1.4; 2.9; 5.8; 29 and 58 µg/mL for moxifloxacin and 2.2; 4.4; 8.8; 44 and 88 µg/mL for vancomycin). The y-axis of panel B corresponds to the y-axis of panel A. The experiments were performed in triplicates and each value is presented as the mean ± standard deviation (n = 3).

5. Figure 6- all experiments supposedly have an n=3, but some appear to only have 1 or 2 graphed (i.e. amox clone 1 dilute stationary). If the datapoints overlap is it possible to have them side by side such that all datapoints can be visualized?

Thank you for this attentive comment. Due to experimental issues in some conditions, we indeed are missing some datapoints. As this does not really affect the point we want to make (namely, that survivors are generally not different from the original clone in survival levels), we opted to analyze the dataset with a mixed effect model.

6. I think the y-axis in Figure 7 is incorrect or is not to scale. The scale goes from 1 to 0.01 to 0.001. Is the 1 supposed to be 0.1?

Thank you for noticing this error in the y-axis from Figure 7. We adjusted the scale to the correct values ($10^2 - 10^0 - 10^{-2} - 10^{-4} - 10^{-6}$).

Figure 7

Major Comments

1. While of interest, the inclusion of the *lytA/spxB* detracts from the overall narrative. In my opinion, while these efforts were smart avenues to address the question at hand, the utilization of the alternative media that was used for the remainder of the manuscript is where the bulk of the experimental data is focuses. Cutting these sections would make the manuscript more concise and focused. To this end, I might revise figure 1 to indicate the Thy and MHL results. The *lytA/spxB* data could be placed in supplemental data. Figure 9 could also be moved to supplemental data.

As per your suggestions, we moved the data with *lytA/spxB* from the main text and included the data in supplemental data figure 1 to make the text more focused. We also moved figure 9 to the supplemental data section.

Figure 1

Figure 1: MHL abolishes the self-limiting *in vitro* nature of *S. pneumoniae*.

We compared the planktonic growth curves of *S. pneumoniae* in BHI (Brain-Heart Infusion broth), THY (Todd-Hewitt broth supplemented with 0.5% Yeast extract) and MHL (Mueller-Hinton broth supplemented with 5% Lysed horse blood). Survival is higher when *S. pneumoniae* is grown in MHL than in THY or BHI. The experiments were performed in triplicates and each value is presented as the mean \pm standard deviation ($n = 3$).

Figure S1

Figure S1: The self-limiting *in vitro* nature of *S. pneumoniae* is not counteracted in BHI and THY by different strategies adopted to avoid H₂O₂ or autolysis.

We compared the effect of different strategies to counteract the effects of pyruvate oxidase (A, B and C) or autolysin (D, E and F) in planktonic growth curves of *S. pneumoniae* D39 in BHI (Brain Heart Infusion broth, A and D), THY (Todd-Hewitt broth supplemented with 0.5% Yeast extract, B and E) and MHL (Mueller-Hinton broth supplemented with 5% Lysed horse blood, C and F). The strong reduction of viable bacteria after 8 hours of growth is still observed despite the adopted strategies in the media BHI and THY, but MHL abolishes the self-limiting *in vitro* nature of *S. pneumoniae*. The experiments were performed in duplicates or triplicates and each value is presented as the mean \pm standard deviation ($n \geq 2$).

2. I thought the data shown in Figure 7 to be very compelling as the kill kinetics really show some interesting differences. What these experiments lack is two (fairly minor) aspects. First, the number of bacteria from the 1:100 dilution (starting concentration) should be indicated here. Are they all from the same starting point or do they vary considerably in their starting CFU? Another aspect to consider would be to measure growth rate (doubling times) from these 1:100 dilutions of the strains. Are the strains that outgrow following dilution the slowest the most likely to survive? And vice versa, are the once that outgrow the fastest the more susceptible? Even if this does not correlate with the survival, this would be substantially strengthen the findings of this paper.

Thank you for your appraisal of our data regarding the differences in kill kinetics.

First, we would like to clarify the confusion about the dilution factors and thus the starting concentrations. For diluted stationary phase conditions, we diluted a stationary phase culture **1:10** in fresh medium right before the start of treatment (top two panels in Fig. 7). We obtained exponential phase conditions by a 1:200 dilution of a stationary phase culture in fresh medium, to a concentration of approximately 5×10^5 CFU/mL, followed by 3 hours of growth before antibiotic treatment was started (bottom two panels in Fig. 7). Thus, to answer your first point, starting concentrations were in the range from $5 \times 10^6 - 8 \times 10^7$ CFU/mL for the diluted stationary phase and in the range of $2 \times 10^5 - 1 \times 10^7$ CFU/mL for the exponential cultures. We changed the caption of the figure to clarify how we obtained cultures from the different growth phases and we added the ranges of starting concentrations.

Figure legend 7

Figure 7: Antibiotic-tolerant cells are widely present after antibiotic treatment in, and highly variable among *S. pneumoniae* strains from different sources.

*Survival is highly variable in *S. pneumoniae* strains after challenge with 100-fold the MIC of amoxicillin (amox) or of vancomycin (van). An 8-hours antibiotic treatment was started after a 1:10 dilution of stationary phase bacteria (Dil stat) or after dilution of stationary phase bacteria to 5×10^5 CFU/mL followed by 3 hours of growth (exponentially growing bacteria; Exp). Starting concentrations were in the range from $5 \times 10^6 - 8 \times 10^7$ CFU/mL for the diluted stationary phase and in the range of $2 \times 10^5 - 1 \times 10^7$ CFU/mL for the exponential cultures. Strains are ordered based on their survival after treatment with amoxicillin in the diluted stationary phase. Log-transformed data are shown as a boxplot of the mean \pm standard deviation for each strain. The experiments were performed in triplicates and each individual measurement is given as a dot according to the repeat ($n = 3$).*

We now also have included additional correlation analyses in supplemental data figure 3. Here, we test for correlations between the initial CFU before treatment and the survival, which indicates a small negative correlation. When antibiotic treatment was initiated at a culture with a lower bacterial concentration, in both growth phases, survival after antibiotic treatment was higher. We added the additional analysis to supplementary data figure 3 and we discussed the additional correlation analysis in the main text. Finally, we would like to thank you for your suggestion regarding the correlation between survival and growth. We performed and discussed the correlation analysis between survival and growth, which showed a small positive correlation between increased growth and increased survival.

Lines 263-271

Survival levels correlated strongly between the growth phases (diluted stationary and exponential) and the antibiotics (amoxicillin and vancomycin) (Figure 8 and Figure S3). Surprisingly, we also detected a small positive correlation (Pearson correlation, R^2 ranging from 0.032 – 0.26, $p \leq 0.05$ for

all conditions, except for vancomycin in the diluted stationary phase) between increased growth and increased survival, which does not substantiate the common belief that slow growth induces persister formation (53–59). Finally, we detected a small negative correlation between initial CFU before treatment and survival (Pearson correlation, R^2 ranging from $2.1 \cdot 10^{-6}$ – 0.11, $p \leq 0.05$ for amoxicillin in the exponential phase) implicating that survival was higher when antibiotic treatment was started on a culture with a lower bacterial concentration (Figure S3).

Figure S3

Figure S3: Correlation analysis of survival fractions between different conditions show strong correlations between antibiotics (amoxicillin and vancomycin) and growth phases (diluted stationary and exponential growth phase).

Individual correlations between the survival rates in 4 different conditions: treatment with amoxicillin (amox) or vancomycin (van) in the diluted stationary (dilstat) or exponential (exp) growth phase. In addition, the correlation with the corresponding control (growth in absence of antibiotics during the period of treatment in exponential or diluted stationary phase conditions) and with the initial CFU before treatment (pre treat) is given. Pearson correlation coefficients (R^2) are given for each correlation. A.u., arbitrary units.

Reviewer #2 (Comments for the Author)

In the manuscript entitled, "Antibiotic-tolerant persisters are pervasive among clinical *Streptococcus pneumoniae* isolates and show strong condition-dependence", Geerts et al provide evidence for the presence of persister cells in *Streptococcus pneumoniae* D39 strain following antibiotic treatment. There were two supporting pieces of evidence - first, the bacterial population showed the characteristic biphasic killing pattern, and second, there was mostly no change in the MIC of persister cells relative to rest of the sensitive bacterial population. The number of persister cells varied depending on the growth phase of the bacteria, and the type of antibiotic used. While the authors attempt to demonstrate that this phenomenon is well conserved across different pneumococcal strains, they have not provided sufficient evidence to support this claim (more below). While this is an important study on a clinically relevant subject, the manuscript is missing some controls to convincingly report the persistence of persisters in *S. pneumoniae*. Additionally, the manuscript will be more robust if certain statements and overinterpretations are toned down.

We would like to thank you for your time to review the manuscript and for your critical view. We believe that the revised manuscript has substantially improved upon taking your comments into account.

Major Comments

1. Lines 169 - 173:

a. Authors state that significant killing of sensitive cells occurred at or above a concentration of 20X MIC. Did concentrations below that threshold not result in sufficient killing of sensitive cells in the observed time frame? Please provide data for the same. Further, it is unclear why a concentration of 100X the MIC was chosen for rest of the work - please elaborate on the rationale. For instance, why not 20X or 50X the MIC?

We performed additional experiments to assess survival after a 5 hours treatment with 5X and 10X the MIC of the different antibiotics. We included an untreated control and treatment with 100X the MIC. The results are shown in the figure below and in the revised figure 4. As can be seen, the plateau in survival starts at 5X the MIC and, theoretically, lower concentrations than 100X the MIC could be sufficient for the strain that was used in this experiment. However, using an excess concentration is common in persistence research (1–5) and generally serves multiple purposes beyond ensuring proper killing. For one, small variations are less important when working with concentrations much higher than the threshold. Second, the MIC of single-step mutants generally does not quickly approach the higher doses used here, so resistance is less likely to affect our conclusions. Third, lower concentrations could lead to slower killing of normal cells, resulting in longer treatment times to reach the persister plateau in function of time. Lastly, when working with various additional conditions and strains, a higher treatment concentration could level out small differences in threshold concentrations that are needed to reach a plateau between the conditions. In the end, one needs to make a choice and we cannot perform an optimization for all the conditions/strains that were used.

Figure 4

b. A hallmark of persistence is that the size of the persister subpopulation is only weakly dependent on the antibiotic concentration. Is that true for this work? The authors have only showed proportion of persister cells formed at 100X the MIC, and not any other concentration.

Generally speaking, we would indeed expect little dependency towards concentration once a sufficient dose is reached to kill sensitive cells. Here, to limit the number of variables to be tested and keep the focus of the article, we only checked concentration dependency in Fig 4. As 5h treatment is sufficient to reach the second phase of the killing, we assume that surviving cells after treatment with 20X or 200X the MIC are persisters as we could not observe any concentration dependence. However, also in light of some of your other comments, we nuanced our reasoning, so we don't claim that all surviving cells are persisters, especially in the later parts of the manuscript.

Lines 169-176

The independence on antibiotic concentration, once a sufficient dose is reached to kill sensitive cells, is a typical observation indicating a role for persistence while strong correlations would point towards antibiotic resistance as underlying cause of survival (4). Here, the independence on antibiotic concentration could be the first indication of the presence of persister cells within S. pneumoniae D39 cultures. For the remainder of our work, we applied concentrations of 100-fold the MIC to ensure proper killing of sensitive cells and because a lower antibiotic concentration could lead to slower killing of normal cells, which would result in longer treatment times needed to reach the persister plateau in function of time.

2. Lines 195-197: The reported proportion of persister cells (13.74-60.08%) in dilute stationary phase cultures upon treatment with different antibiotics fall in a wide range, and perhaps needs another look. Since killing data for cefuroxime and vancomycin does not fit biphasic pattern better than uniphasic pattern, it is unclear whether the tolerant cells in these conditions are bonafide persisters. As such, it might be better to report this proportion only when the killing curves show a biphasic pattern i.e. upon treatment with amoxicillin and moxifloxacin.

We have adjusted our wording at this section to accommodate your concern. However, we want to stress that absence of significance is no significance of absence and that test statistics for these two exceptions are actually close to significance (AIC and LogLik are even in favor of the biphasic model).

Lines 193-202

When each condition (growth phase x antibiotic) is analyzed separately, the biphasic model is significantly preferred over the uniphasic model for describing the data from all conditions ($p \leq 0.05$), except for data from treatment with cefuroxime and vancomycin in the diluted stationary growth phase. While this might indicate that including a second killing rate does not improve the models for

these conditions, p values are close to significance ($p = 0.1641$ and 0.1074 , respectively) and various test statistics (AIC, BIC and LogLik) are either inconclusive or in favor of the biphasic model (Table S2). Overall, we detected relatively high persister levels, when the biphasic model is preferred over the uniphasic model, ranging from 13.74 to 24.31%, for amoxicillin and moxifloxacin the diluted stationary growth phase compared to the lower levels, ranging from 0.02 to 0.5%, in the exponential growth phase.

3. Figure 6 and lines 208-212: It is unclear why the authors have used a different time point for comparing the proportion of cells that survive following antibiotic treatment when grown in exponential phase. It seems that for initial antibiotic treatment, cells were treated with antibiotic for 18 hours. But in the subsequent round, the corresponding time point was 6 hours. Why is that?

We thank the reviewer for pointing out this inconsistency. We initially chose the 6 hours treatment for the heritability assay for practical reasons, but given the difference among the two conditions in killing dynamics, following the treatment duration of the initial experiment makes more sense. We therefore repeated the experiment with an 18-hours treatment for the exponentially growing cultures and show the updated data in the revised figure 6.

Figure 6

Figure 6: **The antibiotic tolerance of surviving *S. pneumoniae* cells is transient and non-deterministically inherited by daughter cells.**

AB-tolerant *S. pneumoniae* D39 clones were recovered after 6 (Dil stat) or 18 (Exp) hours of treatment during the initial time-kill assay, regrown without antibiotic and preserved at -80°C . On these clones arising from potential persister cells, survival was determined after 6 (Dil stat) or 18 (Exp) hours of antibiotic treatment with amoxicillin (amox), cefuroxime (cef), moxifloxacin (mox) and vancomycin (van) in the diluted stationary (Dil stat) or the exponential growth phase (Exp). Survival of the randomly selected clones was similar to the original culture (mixed-effect analysis, clone 1 was excluded from the analysis for amox – Dil stat, because we had only one datapoint). The experiments were performed in duplicates or triplicates and each value is presented as the mean \pm standard deviation ($n \geq 2$).

Additionally, in Fig 6 - for diluted stationary phase cultures, most antibiotic treatments (except for vancomycin) have 2 data points each for persister clones. It is also unclear how ANOVA was performed with <3 data points.

We performed three biological repeats of the experiment, but with some repeats we encountered technical issues resulting in missing datapoints. Unfortunately, our statistical program did not raise a flag. However, we feel that this does not really affect the point we want to make (namely, that survivors are generally not different from the original clone in survival levels) and we opted to analyze the dataset with a mixed effect model. Clone 1 was not included in the analysis for amoxicillin in the diluted stationary phase, because we had only one datapoint.

4. Lines 216- 246 (and other places): The authors have shown that there is some survival of cells when additional strains and clinical isolates are treated with different antibiotics. However, they have not demonstrated that these surviving cells are persisters for the following reasons -
a. Survival was reported for a time point as early as 3 hours following antibiotic treatment of exponentially growing bacteria. Even for D39 strain, at such an early time point (eg Fig 5) - the cells being killed are sensitive to antibiotics and not persisters. Thus, reporting these surviving cells as persisters (from data in Fig 7) seems misleading and inaccurate.

We clarify the confusion regarding the duration of treatment as we did in fact used longer treatments based on the killing curves of the D39 strain. First, bacteria were brought into the right growth phase, diluted stationary or exponential growth phase, after which an antibiotic treatment of 8 hours was started, which lies in the second phase of all killing curves of the D39 strain. We have adapted the text in the figure legend to make the protocol more clear.

Figure legend 7

Figure 7: Antibiotic-tolerant cells are widely present after antibiotic treatment in, and highly variable among *S. pneumoniae* strains from different sources.

*Survival is highly variable in *S. pneumoniae* strains after challenge with 100-fold the MIC of amoxicillin (amox) or of vancomycin (van). An 8-hours antibiotic treatment was started after a 1:10 dilution of stationary phase bacteria (Dil stat) or after dilution of stationary phase bacteria to 5×10^5 CFU/mL followed by 3 hours of growth (exponentially growing bacteria; Exp). Starting concentrations were in the range from 5×10^6 – 8×10^7 CFU/mL for the diluted stationary phase and in the range of 2×10^5 – 1×10^7 CFU/mL for the exponential cultures. Strains are ordered based on their survival after treatment with amoxicillin in the diluted stationary phase. Log-transformed data are shown as a boxplot of the mean \pm standard deviation for each strain. The experiments were performed in triplicates and each individual measurement is given as a dot according to the repeat ($n = 3$).*

b. No data regarding the kinetics of antibiotic-induced killing has been shown. Is the killing pattern uniphasic or biphasic?

See response to 4.c.

c. Is there any change in the MIC of the surviving cells? Is the survivability phenotype heritable? Without this data, no conclusions can be drawn on whether these surviving cells are persisters or not.

We understand the rationale behind the last two comments. However, the aim of this study was to optimize persistence assessment with extensive experimentation using the reference strain D39 after which we used the optimized setup during a screening of clinical isolates. Given the large number of

tests in the latter screening, it is impossible in the current study to go into the same depth as was done for the reference D39 strain (e.g. biphasic killing dynamics, checking surviving colonies). Therefore, we can only assume that large differences in survival levels that we report result from differences in persistence among the tested *S. pneumoniae* strains. Despite the indications from the optimization with the D39 strain that surviving cells are persisters, it remains possible that surviving cells of isolates are resistant mutants (but see our previous comment on the high antibiotic dose that was used, we deem it unlikely that these sensitive strains became resistant during a single antibiotic treatment of 8 hours at 100X the MIC) or that killing dynamics are strongly different in some of the clinical strains so that survival levels on one timepoint do not correctly reflect differences in persistence levels. We made nuances in the text to make it clear that we assume surviving cells to be persisters rather than calling them persisters.

Lines 222-224

*Having established the presence of persisters in strain D39 in an optimized set-up, we wondered whether other *S. pneumoniae* strains could survive antibiotic treatment, assumingly through the presence of persister cells, and how this phenotype varies within the *S. pneumoniae* species.*

Lines 251-260

*After the determination of survival levels of clinical isolates in an optimized set-up, we assume that the large differences in survival levels reflect differences in persistence among the tested *S. pneumoniae* strains. Despite the indications from the optimization with reference strain D39 that surviving cells are persisters, it remains however possible that killing dynamics are strongly different in some of the clinical strains so that survival levels on one timepoint do not correctly reflect differences in persistence levels. For example, a difference in killing rates of normal cells and persisters can result in a similar survival level after antibiotic treatment, but with no or a different level of persisters. Since our current work cannot test this more complex explanation, we therefore conclude that survival, likely through persistence, is highly variably in *S. pneumoniae* strains from different sources.*

5. Lines 251-253, 320-325: A Pearson's correlation as small as 0.032 is perhaps more reflective of no correlation than positive correlation. Thus, in the absence of other supporting data to support positive correlation, the conclusion that the data "contradicts the common believe (belief) that slow growth induces persister formation" seems like an overinterpretation.

Thank you for this comment. We rephrased the text saying that our data does not substantiate the common belief.

Lines 265-268

Surprisingly, we also detected a small positive correlation (Pearson correlation, R^2 ranging from 0.032 – 0.26, $p \leq 0.05$ for all conditions, except for vancomycin in the diluted stationary phase) between increased growth and increased survival, which does not substantiate the common belief that slow growth induces persister formation (53–59).

6. Lines 40-41, 331-333 etc: The assertion that a "broad range of genetic elements" are controlling persister generation is speculative. The potential conservation of the phenotype across diverse pneumococcal strains alone does not necessarily indicate that numerous genetic elements or accessory genes are responsible for persister formation. It is possible that a few near core genes are responsible for the phenotype.

We made this assumption based on the knowledge in other bacterial species, for example *E. coli*, for which many genetic markers were found to be involved in persister formation (6–8). We agree with the reviewer that the high diversity among different *S. pneumoniae* cultures could also be explained by only a few core genes (e.g. that differ in expression level). We have omitted our statement from the revised version.

7. Lines 292-293: Could the authors speculate on the reasons behind why persistence differed between growth phases and antibiotics.

We would like to thank you for this suggesting and extended the discussion about the potential causes of differences in persistence levels.

Lines 312-320

Survival levels were vastly higher in diluted stationary phase cultures than for exponentially growing bacteria for all examined strains, as we expected, because persistence is mostly linked to dormancy and bacteria from the diluted stationary phase recently came out the stationary phase and could therefore be less metabolically active (21, 43, 73). The difference in persistence between antibiotics can be attributed to the different modes of action of the antibiotics. Interestingly, treatment of diluted stationary phase cultures with moxifloxacin, a fluoroquinolone that targets the DNA synthesis of bacteria and is less dependent on cell growth than β -lactams, resulted in the lowest level of persisters (13,74%) in diluted stationary phase cultures (47, 74, 75).

8. (Optional): Is there any correlation between the resistance level of a strain (high vs low MIC) and its propensity to form persisters? For instance, strain 85 is resistance to cefuroxime and CI 7 is reported to be resistant to moxifloxacin. Are there any persisters present in these strains following treatment with these respective antibiotics? If so, what is the percent of the population that forms persisters?

Thank you for this interesting suggestion. We analyzed the link between the MIC and the survival fraction for amoxicillin and vancomycin in the figure below. Given the high amount of data already in the manuscript, we opted to only provide this analysis in the rebuttal. There is no clear correlation between MIC and survival fraction for amoxicillin in both growth phases. A negative correlation is observed between the survival fraction and the MIC for vancomycin in both growth phases, which contradicts the logic hypothesis that higher MIC levels would also result in higher survival levels. None of the correlations is significant ($p > 0.05$). Screening the isolates for survival after treatment with cefuroxime and moxifloxacin is out of the scope of this study and assessing persistence in strongly resistant strains is challenging given the potential confounding effect.

Minor Comments

1. Lines 117, 121, and elsewhere. Nomenclature for gene name - the last letter should be italicized (spxB, lytA).

We apologize for these errors and we adjusted the text to the right nomenclature for the gene names.

2. Lines 90-91: While persister formation has not previously been investigated, there are some studies have investigated the antibiotic tolerance phenotype observed in pneumococcus. Since pneumococcus also causes chronic infections (as acknowledged later in the paragraph) and is a common colonizer, the statement that persisters may have been ignored because of the "acute" nature of pneumococcal infections seems misleading.

We concur with this statement. We changed the text so we don't claim that pneumococcal persisters may have been ignored because of the acute nature of pneumococcal infections.

Lines 90-95

Antibiotic-tolerant persisters are mostly connected with recurrent and chronic infections and the role of persisters in acute infections is not clear (7, 18). Most infections caused by S. pneumoniae have an acute nature. Nonetheless, S. pneumoniae is also, albeit to a lesser extent, the causative agent of chronic diseases, like chronic endobronchial infections in children (40–42) and it can reside in biofilms in the middle ear of children causing recurrent and chronic otitis media (43–46).

3. Lines 362-363: Please report how much catalase and choline chloride were added.

We apologize for not providing the used concentrations of catalase and choline chloride. We added this information in the materials and methods section.

Lines 382-383

Bacteria were grown in different media with or without catalase (1000 U/mL) or choline chloride (10 mM) supplementation.

4. Fig 9 - It might be better to include this figure as a supplement figure as opposed to a main figure since it is tangential to the main findings being reported.

We concur with this comment. Figure 9 is moved to the supplemental data section.

5. Typos: Despite "being" frequently (line 55), set "the" stage (line 55), Another important clinical (line 74), round of "antibiotic" treatment (line 206), "specifically" about (line 283)

Thank you for noticing these errors. We corrected the typos in the text.

Reference list

1. Barrett TC, Mok WWK, Murawski AM, Brynildsen MP. 2019. Enhanced antibiotic resistance development from fluoroquinolone persisters after a single exposure to antibiotic. *Nat Commun* 10.
2. Mulcahy LR, Burns JL, Lory S, Lewis K. 2010. Emergence of *Pseudomonas aeruginosa* strains producing high levels of persister cells in patients with cystic fibrosis. *J Bacteriol* 192:6191–6199.
3. Nierman WC, Yu Y, Losada L. 2015. The In vitro Antibiotic Tolerant Persister Population in *Burkholderia pseudomallei* is Altered by Environmental Factors. *Front Microbiol* 6.
4. Willenborg J, Willms D, Bertram R, Goethe R, Valentin-Weigand P. 2014. Characterization of multi-drug tolerant persister cells in *Streptococcus suis*. *BMC Microbiol* 14:120.
5. Jiang YL, Qiu W, Zhou XD, Li H, Lu JZ, Xu HHK, Peng X, Li MY, Feng MY, Cheng L, Ren B. 2017. Quaternary ammonium-induced multidrug tolerant *Streptococcus mutans* persisters elevate cariogenic virulence in vitro. *Int J Oral Sci* 9.
6. Hofsteenge N, Van Nimwegen E, Silander OK. 2013. Quantitative analysis of persister fractions suggests different mechanisms of formation among environmental isolates of *E. coli*. *BMC Microbiol* 13.
7. van den Bergh B, Fauvart M, Michiels J. 2017. Formation, physiology, ecology, evolution and clinical importance of bacterial persisters. *FEMS Microbiol Rev* 41:219–251.
8. Gollan B, Grabe G, Michaux C, Helaine S. 2019. Bacterial Persisters and Infection: Past, Present, and Progressing. *Annu Rev Microbiol* 73:359–385.

October 16, 2022

Prof. Paul Cos
University of Antwerp
Laboratory of Microbiology, Parasitology and Hygiene
Universiteitsplein 1
wilrijk, antwerpen 2610
Belgium

Re: Spectrum02701-22R1 (Antibiotic tolerance indicative for persistence is pervasive among clinical *Streptococcus pneumoniae* isolates and shows strong condition dependence)

Dear Prof. Paul Cos:

Your manuscript has been accepted, and I am forwarding it to the ASM Journals Department for publication. You will be notified when your proofs are ready to be viewed.

Sincerely,

John Atack
Editor, Microbiology Spectrum
